# Outlining the Psychological Profile of Persistent Depression in Fibromyalgia Patients Through Personality Assessment Inventory (PAI)

**DOI:** 10.3390/ejihpe15010002

**Published:** 2025-01-06

**Authors:** Andrea Doreste, Jesus Pujol, Eva Penelo, Víctor Pérez, Laura Blanco-Hinojo, Gerard Martínez-Vilavella, Helena Pardina-Torner, Fabiola Ojeda, Jordi Monfort, Joan Deus

**Affiliations:** 1Department of Clinical and Health Psychology, Autonomous University of Barcelona, 08193 Bellaterra, Spain; 2MRI Research Unit, Radiology Department, Hospital del Mar, 08003 Barcelona, Spain; 21404jpn@comb.cat (J.P.); g.martinezvilavella@gmail.com (G.M.-V.); 3Centro de Investigación Biomédica en Red de Salud Mental (CIBERSAM G21), Instituto de Salud Carlos III, 08036 Barcelona, Spain; vperezsola@psmar.cat; 4Departament de Psicobiologia i de Metodologia de les Ciències de la Salut, Universitat Autònoma de Barcelona, 08193 Bellaterra, Spain; eva.penelo@uab.cat; 5Neurociences Research Unit, IMIM-Institut Hospital del Mar d’Investigacions Mèdiques, 08003 Barcelona, Spain; 6Department of Experimental and Health Sciences, Pompeu Fabra University, 08002 Barcelona, Spain; 7Cognition and Brain Plasticity Unit (Bellvitge Biomedical Research Institute–IDIBELL), 08908 Barcelona, Spain; hpardinatorner@ub.edu; 8Rheumatology Department, Hospital del Mar, 08003 Barcelona, Spainjmonfort@parcdesalutmar.cat (J.M.)

**Keywords:** fibromyalgia, psychopathology, Personality Assessment Inventory (PAI), dysthymia, MCMI-III, major depression

## Abstract

**Background:** Fibromyalgia (FM) is a complex condition marked by increased pain sensitivity and central sensitization. Studies often explore the link between FM and depressive anxiety disorders, but few focus on dysthymia or persistent depressive disorder (PDD), which can be more disabling than major depression (MD). **Objective:** To identify clinical scales and subscales of the Personality Assessment Inventory (PAI) that effectively describe and differentiate the psychological profile of PDD, with or without comorbid MD, in FM patients with PDD previously dimensionally classified by the Millon Clinical Multiaxial Inventory III (MCMI-III). **Method:** An observational, cross-sectional study was conducted with 66 women (mean age 49.18, SD = 8.09) from Hospital del Mar. The PAI, the MCMI-III, and the Fibromyalgia Impact Questionnaire (FIQ) were used to assess the sample. **Results:** The PAI showed strong discriminative ability in detecting PDD, characterized by high scores in cognitive and emotional depression and low scores in identity alteration, dominance, and grandeur. High scores in cognitive, emotional, and physiological depression, identity alteration, cognitive anxiety, and suicidal ideation, along with low scores in dominance and grandeur, were needed to detect MD with PDD. Discriminant analysis could differentiate 69.6–73.9% of the PDD group and 84.6% of the PDD+MD group. Group comparisons showed that 72.2% of patients with an affective disorder by PAI were correctly classified in the MCMI-III affective disorder group, and 70% without affective disorder were correctly classified. **Conclusions:** The PAI effectively identifies PDD in FM patients and detects concurrent MD episodes, aiding in better prognostic and therapeutic guidance.

## 1. Introduction

Fibromyalgia (FM) is a complex disease characterized by heightened pain sensitivity with a multifactorial etiopathogenesis ([54]). It is defined as a central sensitization syndrome ([1]; [2]) due to aberrant central pain processing ([34]; [46]; [52]). Comorbidly, individuals with FM often exhibit psychopathological disorders, affecting 13–80% of cases, which are predominantly categorized within affective spectrum disorders ([3]; [14]; [16]; [31]). Mental health problems seem to play a significant role in the development, course and maintenance of this condition ([16]; [24]), with a dysfunctional psychological coping state considered key to poorer physical health and quality of life ([14]; [16]; [35]; [17]).

FM has been associated with biological etiologies, including central hyperactivation, endocrine system dysfunction, and altered sensory processing ([46]). Additionally, psychological factors play a significant role in the characteristic pain of fibromyalgia ([54]; [16]). Various psychological models provide insights into the interaction between central sensitization, emotional factors, and social stressors. The Biopsychosocial Model illustrates how these elements disproportionately affect affective spectrum disorders in fibromyalgia. Cognitive Behavioral Theory links dysfunctional thought patterns, such as catastrophizing and pain hypervigilance, to increased pain perception and emotional distress. Social Cognitive Theory highlights the impact of low self-efficacy and ineffective coping on functional disability, while the Somatic Symptom Model underscores how psychological factors mediate physical symptom amplification ([16]).

FM patients do not form a clinically homogenous group, and not all of them exhibit comorbid mental disorders ([17]; [15]; [19], [20]). However, there is a specific psychopathological profile associated with FM that distinguishes it from other chronic pain conditions ([21]; [33]). The presence of FM tends to have a more pronounced impact on mood and anxiety disorders ([31]; [17]; [23]; [26]). Among these, major depressive disorder is the most prevalent, affecting 28–63% of women and 23–42% of men, while dysthymic disorder affects 50–53% of FM patients ([31]; [17]; [23]). Nearly one-third of FM patients may also suffer other mental conditions, including bipolar disorder (26%), panic disorder (33%), post-traumatic stress disorder (16–39%), anxiety disorder (8–30%), obsessive–compulsive disorder (2–4%), and specific phobias (14–17%) ([31]). In addition to these psychological issues, FM symptoms such as cognitive performance problems and somatoform problems ([6]), general fatigue, and poor sleep quality ([35]; [4]; [22]) are also associated with psychopathological problems like pain catastrophizing, pain hypervigilance, and lower levels of pain self-efficacy and pain acceptance ([36]).

Several studies have examined the link between depressive anxiety disorders and FM, although few have focused on dysthymia or persistent depressive disorder (PDD), which is often more disabling than major depression (MD) ([51]). PDD is a chronic state of depressed mood for most of the days, lasting 2 years or longer. It implies a high risk of developing MD ([51]; [56]) and particularly showcases a melancholic temperament among those affected patients ([56]). In fact, this affective disorder has often been called “double depression” ([30]), with personality disorders ([17]; [51]) and somatoform conditions ([56]) being commonly comorbid ([51]; [56]). Indeed, following the current Diagnostic and Statistical Manual of Mental Disorders (DSM) criteria, FM patients exhibit a higher prevalence of a diagnosis related to somatic disorder and/or PDD in comparison to other chronic pain conditions without central sensitization syndrome ([33]; [4]; [10]). In the broader population with chronic pain, the prevalence of MD and PDD falls within ranges between 10 and 30% ([25]). Furthermore, only chronic fatigue, frequently associated with FM, displays a higher prevalence of PDD than MD ([26]).

The assessment of affective disorders in FM presents challenges due to inconsistent findings ([32]), attributed to overlapping somatic symptoms ([14]) and varying study criteria ([55]). Nevertheless, identifying depressive disorders, especially PDD, is essential as they cause intense suffering for these patients with particular needs and deficits ([5]). For this purpose, a variety of psychopathological multidimensional screening tools have been utilized, including the Symptom Checklist-90-Revised (SCL-90-R) ([29]; [18]; [49]), the Minnesota Multiphasic Personality Inventory (MMPI) ([19], [20], [21]; [44]; [42]), the Millon Clinical Multiaxial Inventory (MCMI) ([17]; [33]; [6]; [42]), and recently the Personality Assessment Inventory (PAI) ([13]). The MCMI shows appealing measurement characteristics for routine clinical practice, particularly its brief administration and robust theoretical focus based on dimensional criteria and classification approximation to the DSM, facilitating the assessment of psychopathological and personality issues ([6]; [48]; [53]). Specifically, it demonstrates diagnosis efficiency for both categorical and dimensional assessment of PDD and melancholic personality traits ([50]), distinguishing itself from other screening or multidimensional tests.

The PAI is effective in uncovering a range of psychopathological and personality concerns, boasting robust psychometric properties, particularly among individuals with chronic pain ([28]), including those with FM ([13]). Furthermore, PAI allows for more specific identification of affective disorder subtypes and other clinical disorders, better assessment of personality disorders, and a reduced emphasis on psychosomatic issues. However, the PAI may not fully capture the psychopathological profile associated with PDD, a limitation that is addressed by using the MCMI. This study aims to examine which clinical scales and subscales scores of the PAI can accurately describe and effectively discriminate the psychological profile of PDD, with or without MD, in a sample of FM patients previously classified based on the MCMI criterion for the PDD profile.

## 2. Method

### 2.1. Eligibility Criteria

The research included females aged 18 to 65 diagnosed with FM according to the criteria established by the American College of Rheumatology ([58]). Inclusion criteria involved having a stable pharmacological treatment, understanding the study requirements, and a commitment to compliance. Exclusion criteria encompassed the presence of other conditions explaining pain; inflammatory or rheumatic diseases; severe or unstable medical, endocrine, or neurological conditions; a history of neuropathic pain; acute psychotic disorders; substance abuse; and invalidating scores on the MCMI and PAI validity scales, which could compromise the interpretation of results.

### 2.2. Participants

Patients were enlisted from the Fibromyalgia Unit at Barcelona’s Hospital del Mar by senior rheumatologists (FO or JM) and a senior psychologist (JD) between January 2021 and June 2022. During the study period, 136 female patients were diagnosed with FM, and 110 underwent eligibility assessments during consecutive clinical visits. Forty-four either did not meet the study criteria or declined participation, resulting in a final sample of sixty-six participants who completed both the MCMI-III and PAI questionnaires. Detailed sociodemographic and clinical characteristics can be found in Table 1.

### 2.3. Study Design and Procedure

We used a non-randomized, purposeful sampling method to include all eligible participants from the study population. This was an observational, cross-sectional study. Female patients initially attended their regular rheumatology appointments (FO and JM). After thorough screening for inclusion/exclusion criteria and willingness to participate, they were enrolled in the study, and informed consent was obtained. Psychological assessments, conducted by another senior clinical psychologist (AD), occurred within the same week and lasted up to an hour and a half to prevent response fatigue.

### 2.4. Instruments

**Millon Clinical Multiaxial Inventory III (MCMI-III)**, we administered the Spanish version of the MCMI-III ([9]). The MCMI-III ([38]) is a self-administered questionnaire of 175 true–false items that provide insights into personality and psychopathology patterns aligned with the disorders outlined in the DSM (-IV, -IV-R and -V). It comprises a total of 28 scales: 11 basic personality scales and 3 severe personality pathology scales, 7 clinical syndromes and 3 severe clinical syndromes, 3 modifying indices, and a validation scale. The MCMI-III uses “Base Rate” scores for reporting and interpretation. The Base Rate scores (BR) are classified as follows: general population (0–34), low range (35–59), likelihood of presenting the syndrome/trait (60–74), presence of the syndrome/trait (75–84), severe presence of syndrome/trait (85–115). The last two categories are the cut-off points for the instrument. The Spanish adaptation [44] has shown internal consistency > 0.80 with coefficients in personality scales (0.66–0.89), clinical syndromes (0.71–0.90), test-retest values (0.84–0.96), and sensitivity (0.44–0.92) ([37]; [45]). The MCMI-III’s dimensional and classificatory approach to the DSM makes it a widely used multidimensional inventory for clinicians, serving as a tool to check the reliability and validity of other test scores ([48]). Specifically, the Depressive Personality, Major Depression, and Dysthymia scale scores capture depressive pathology and show large convergent correlations with the depression symptom measure by the Beck Depression Inventory (r = 0.48; r = 0.69; r = 0.67 with *p* < 0.001, respectively). Also, Dysthymia and Major Depression scale scores (at a cut-off of 75) showed an acceptable diagnosis accuracy (Specificity = 0.64; Specificity = 0.87, respectively; both *p* < 0.001) ([50]).

**Personality Assessment Inventory (PAI)** ([40]), in its Spanish version ([8]), is a widely utilized self-administered tool consisting of 344 items designed to assess various psychopathological features and personality disorders. The PAI includes 27 scales encompassing 4 validity scales, 5 complementary validity scales, 11 clinical scales, 5 treatment consideration scales, and 2 interpersonal scales, along with 30 conceptually driven clinical subscales. This comprehensive set of scales and subscales allows for the identification of diverse psychopathological profiles, including 15 clinical syndromes and 11 personality disorders ([45]). Participants rate the 344 items on a Likert-type scale (ranging from 1—not at all true—to 4—very true), and raw scores are converted to T-scores based on Spanish norms. Generally, a T-score > 61 suggests a moderate to marked tendency of a prominent psychopathological trait ([28]; [8]; [41]). Nevertheless, it is important to highlight that certain scales may require distinct cut-off points to achieve discriminative capacity, as outlined in the PAI implementation and interpretation manual ([41]) (Figure 1). The Spanish adaptation exhibited appropriate reliability (0.82), internal consistency (0.78 in the healthy sample and 0.83 in the clinical sample), and satisfactory content and convergent validity when evaluating personality and psychopathology across normative, college, and clinical samples ([8]). In chronic pain patients, the internal consistency reliability coefficients for the PAI full scales and subscales scores are acceptable, consistent with previous reports in chronic pain patients. Consequently, the PAI demonstrated crucial psychometric properties when applied to chronic pain settings ([28]).

**Fibromyalgia Impact Questionnaire (FIQ)** ([7]), administered in its Spanish version ([47]), is a self-reported instrument designed to assess the overall functional impact of FM on an individual’s daily life. It includes a range of aspects, such as physical functioning, work-related difficulties, and psychological distress, providing a comprehensive evaluation of the condition’s effects. Scores range from 0 to 100, where higher values indicate a more pronounced impact. The Spanish version of FIQ exhibits adequate internal consistency (α of 0.81), test-retest reliability after 7 days (significant correlations ranging from 0.52 for fatigue and 0.53 for pain to 0.91 for depression), and internal validity, as well as sensitivity to change ([39]). By capturing the multidimensional impact of FM, we consider that the FIQ serves as a valuable tool for clinicians and researchers to understand the extent of impairment and guide treatment decisions.

### 2.5. Data Analysis

A descriptive analysis of sociodemographic and clinical features was conducted to delineate the characteristics of the entire study sample, utilizing IBM SPSS software (Version 21.0, IBM Corp, Armonk, NY, USA) for all analyses. Also, the study sample was divided into three groups based on the dimensional psychological assessment of the MCMI-III regarding the absence, presence, and type of affective disorder, defining: (1) a group with only PDD (group PDD) characterized by BR scores above 75 on that scale; (2) a group with both PDD and MD (group PDD+MD) with scores exceeding 75 BR scores on both scales; and (3) a group without affective disorder (group NAD), with BR scores below 75 on the dysthymia and major depression scale. Furthermore, we obtained a detailed sociodemographic and clinical characteristic for each group and compared them through the non-parametric Kruskal–Wallis test and Chi-square. Finner’s correction was applied. Statistical significance was set at 5%. So, the data analysis involved three main steps:

**PAI group comparison.** A comparison between the three groups of patients (NAD vs. PDD vs. PDD+MD) was carried out to study the psychopathological profile of the PAI clinical scales and subscales using a non-parametric Kruskal–Wallis test. Finner’s correction was applied. All reported *p*-values were two-tailed.

**Predictors selection.** Pairwise comparisons between each pair of groups NAD, PDD, and PDD+MD were conducted using Dunn’s non-parametric pairwise test. Cohen’s d ([11]) was calculated for each comparison to assess the effect size, with values of 0.20 indicating a small effect, 0.50 medium, and 0.80 large. This analytical approach allows for a more comprehensive exploration of which clinical scales and subscales of the PAI enable the definition of a psychopathological profile consistent with PDD, with and without MD.

**Multiple models.** Two discriminant analyses, using the Wilks’ lambda, were performed to assess whether the set of clinical scales and subscales and complementary items of the PAI adequately discriminated which patients exhibited symptomatology of affective disturbance consistent with PDD, with or without MD (PDD vs. NAD and PDD vs. PDD+MD), using a cut-off point (the centroid). The matrices of homogeneity were tested using Box’s M test of equality of covariance. Canonical correlation was applied to measure the association between the discriminant function and the group of PAI variables. Following this, classificatory analysis and cross-validation demonstrated the allocation accuracy for each discriminant analysis. Once the PAI scales were determined in each group, the PDD, NAD, and PDD+MD groups were reformed using these scales and subscales with a cut-off point of 60, which determines the limit of scores within normality. This allowed for observing the percentage of patients in each group, corroborating the discriminant analysis, and conducting a Kappa analysis to compare the 3 groups according to MCMI-III criteria with those based on the PAI scales.

## 3. Results

After descriptive analysis, comparison analysis was applied to the groups with or without affective disorder, such as PDD without MD (PDD; 34.8%; n = 23), with MD (PPD+MD; 19.7%; n = 13), and NAD (NAD; 45.5%; n = 30). Patients in the PDD+MD group showed significantly higher severity of fatigue and stiffness after resting (*p* = 0.034 and *p* = 0.017, respectively) and mood symptoms (depression *p* = 0.012 and anxiety *p* < 0.005). Additionally, these patients exhibited a significantly worse overall Fibromyalgia Impact Questionnaire (FIQ) score (*p* = 0.028), with increased morning tiredness and depression (*p* = 0.034 both) (Table 2).

**PAI group comparison.** We compared the psychopathological profile on the PAI among FM groups as per the MCMI-III criteria (Figure 1A,B). All three groups exhibited significantly high T-scores, with mean scores exceeding the defined cut-off point, on the clinical scales of somatization, anxiety, and depression, but only the PDD+MD group had high scores on the anxiety-related disorders scale, schizophrenia scale, and on a single treatment scale, corresponding to suicidal ideation. Moreover, the PDD+MD group exhibited a statistically significant greater elevation on three clinical scales and one treatment scale: anxiety (M = 73.85, SD = 0.38), depression (M = 81.92, SD = 6.30), schizophrenia (M = 65.31, SD = 8.10), and increased suicidal ideation (M = 70.00, SD = 16.88). The PDD group also showed high scores in anxiety (M = 66.09, SD = 10.48) and depression (M = 73.78, SD = 6.37). Additionally, the PDD+MD group exhibited values within PAI criteria normality but statistically significantly higher when compared to NAD patients in potential suicide index (M = 73.62, SD = 6.66), potential violence index (M = 58.62, SD = 14.22), and treatment difficulty index (M = 61.23, SD = 10.75); and PDD patients showed statistically significant lower scores in interpersonal scale dominance (M = 44.09, SD = 11.14) (Figure 1A).
Figure 1(**A**) Comparison between groups in PAI scales and complementary items; (**B**) Comparison between groups in PAI subscales. (**A**) The green zone indicates the ranges of normality, according to the psychometric criteria of the PAI. **SOM:** Somatic Complaints; **ANS:** Anxiety; **TRA:** Disorders Related to Anxiety; **DEP:** Depression; **MAN:** Mania; **PAR:** Paranoia; **ESQ:** Schizophrenia; **LIM:** Limit Traits; **ANT:** Antisocial Traits; **ALC:** Problems with alcohol; **DRG:** Problems with drugs; **AGR:** Aggression: **SUI:** Suicidal Ideation; **EST:** Stress; **FAS:** Lack of social support; **RTR:** Refusal to treatment; **DOM:** Dominance; **AFA:** Affability; **SIM:** Simulation Index; **DEF:** Defensiveness Index; **IPS:** Potential Suicide Index; **IPV:** Potential index of violence; **IDT:** Treatment Difficulty Index; **FM:** fibromyalgia group; **GC:** theoretical control group; (**B**). Note: **SOM-C:** Conversion; **SOM-S:** Somatization; **SOM-H:** Hypochondria; **ANS-C:** Cognitive; **ANS-E:** Emotional; **ANS-F:** Physiological; **TRA-O:** Obsessive-compulsive; **TRA-F:** Phobias; **TRA-E:** Posttraumatic Stress; **DEP-C:** Cognitive; **DEP-E:** Emotional; **DEP-F:** Physiological; **MAN-A:** Activity level; **MAN-G:** Grandeur; **MAN-I:** Irritability; **PAR-H:** Hypervigilance; **PAR-P:** Persecution; **PAR-R:** Resentment; **ESQ-P:** Experiences. Psychotics; **ESQ-S:** Social Indifference; **ESQ-A:** Alteration of the Thought; **LIM-E:** Emotional instability; **LIM-I:** Alteration of identity; **LIM-P:** Problematic Interpersonal Relationships; **LIM-A:** Self-aggression; **ANT-A:** Antisocial Behaviours; **ANT-E:** Egocentrism; **ANT-B:** Search for sensations; **AGR-A:** Attitude aggressive; **AGR-V:** Verbal aggression; **AGR-F:** Physical assaults; * statistically significant differences between groups; * *p* < 0.05; ** *p* < 0.01.
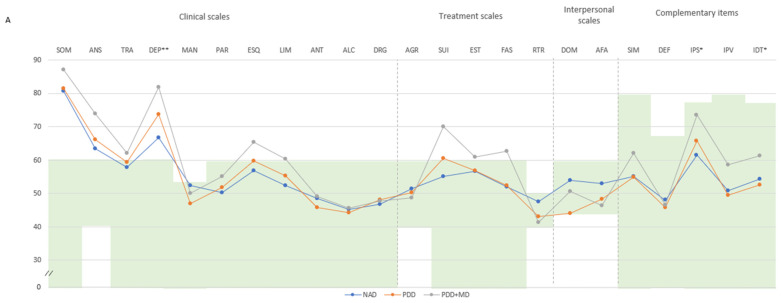

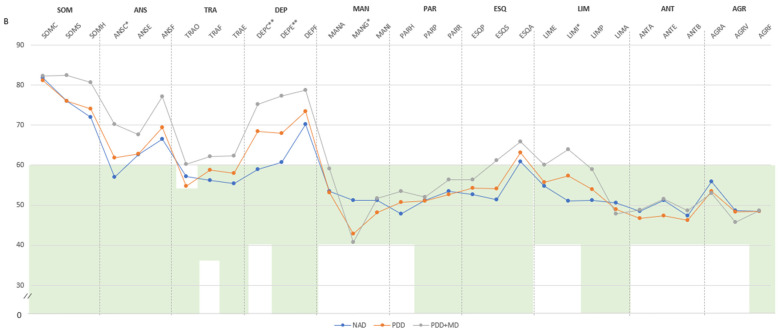



According to the clinical subscales (Figure 1B), all three groups of FM patients exhibited high T-scores in illness-health concern, cognitive anxiety, physiological anxiety, cognitive depression, emotional depression, physiological depression, and thought alteration. Nevertheless, the PDD+MD group showed statistically significant elevations compared to those with NAD and PDD in illness-health concern (M = 80.62, SD = 7.84), cognitive anxiety (M = 70.15, SD = 10.89), physiological anxiety (M = 77.08, SD = 12.12), cognitive depression (M = 75.16, SD = 8.10), emotional depression (M = 77.23, SD = 11.60), and physiological depression (M = 78.69, SD = 6.67). The PDD group also scored high in illness-health concern (M = 74.04, SD = 10.68), cognitive anxiety (M = 61.83, SD = 10.59), physiological anxiety (M = 69.35, SD = 12.87), cognitive depression (M = 68.35, SD = 9.57), emotional depression (M = 68.00, SD = 8.37), and physiological depression (M = 73.48, SD = 7.61), and the NAD group also scored high in illness-health concern (M = 71.97, SD = 11.06), physiological anxiety (M = 66.57, SD = 9.76), and physiological depression (M = 70.13, SD = 8.32). Additionally, the PDD group exhibited a score within PAI criteria normality but statistically significantly lower in grandeur (M = 42.78, SD = 7.17) and PDD+MD in grandeur (M = 40.77, SD = 8.32); this last group also scored higher in social indifference (M = 61.15, SD = 14.34), thought alteration (M = 65.92, SD = 8.60), and identity alteration (M = 63.92, SD = 7.48) (Figure 1B).

**Predictors selection.** Pairwise comparisons among the three groups were conducted on the statistically significant and most relevant scale scores of PAI identified in the first step, obtaining the Cohen’s d for a further explanation of the data (Table 3). Based on these results, we found a psychopathological profile of the PDD group derived from the PAI scores that is presumed to hinge on two stringent conditions, supported by a moderate to large effect size according to Cohen’s d: (a) large effect size (d ≥ 0.80) for scales showing mean scores exceeding the defined cut-off point (i.e., depression and cognitive depression) and for scales showing mean scores below the cut-off point (dominance); (b) medium effect size (d = 0.50–0.79) for the scales with mean scores surpassing the defined cut-off point (emotional depression), as well as for scales with mean scores below the cut-off point (grandeur and identity alteration). Moreover, from these findings, we identified a psychopathological profile of the PDD+MD group that appears to be contingent on the same procedure but different scales according to Cohen’s d: (a) large effect size (d ≥ 0.80) for those scales showing mean scores exceeding the defined cut-off point (depression and emotional depression) and for those showing mean scores below the cut-off point (suicidal potential index, violence potential index, and treatment difficulty index); (b) medium effect size (d = 0.50–0.79) for the scales with mean scores surpassing the defined cut-off point (anxiety, schizophrenia, suicidal ideation, illness-health concern, cognitive anxiety, physiological anxiety, cognitive depression, physiological depression, and identity alteration).

**Multiple models.** The homogeneity values of the variance-covariance matrices were *p* = 0.428 for the NAD vs. PDD analysis and *p* = 0.571 for the PDD vs. PDD+MD analysis. The standardized coefficients and canonical correlation values of each discriminant analysis are reported in Table 4. First, the centroid values for classifying the PDD group were 0.689 and −0.529 for the NAD group. Subsequently, the centroid values for classifying the groups were −0.535 and 0.946 for the PDD and PDD+MD groups, respectively. The classification accuracy was tested, revealing that 73.9% of the cases were correctly classified into the original PDD group and 66.7% into the NAD group. Finally, cross-validation classification resulted in 69.6% and 84.6% for PDD and PDD+MD groups, respectively. Subsequently, the PDD, NAD, and PDD+MD groups were redefined using these scales and subscales, resulting in 28.8% classified as dysthymia by PAI (PDD2), 47% no-affective disorder by PAI (NAD2), and 24.2% dysthymia with major depression by PAI (PDD+MD2). The scales and subscales extracted from this outcome discriminated between 84.2 and 94.7% of PDD2 and 100% of PDD+MD2 cases. Additionally, a Kappa analysis comparing the groups defined by MCMI-III and the groups derived from the PAI scales and subscales showed a match of 43.5% in PDD vs. PDD2 and 53.8% in PDD+MD vs. PDD+MD2, with an overall agreement of 72.2% for affective disorders and 70% for NAD vs. NAD2.

## 4. Discussion

The primary aim of this research was to delineate two psychological profiles based on the PAI that will facilitate the detection of persistent depressive disorder (PDD) or dysthymia and its co-occurrence with major depression (MD) in FM patients. Comparative analysis between groups allowed us to define a psychological profile by PAI of pure PDD, characterized by normal or low scores in dominance, grandeur, and identity alteration and high scores in cognitive and emotional depression. Additionally, we defined a psychological profile of PDD concurrent with MD, along with higher scores in cognitive, emotional, and physiological depression, cognitive anxiety, identity alteration, and suicide potential index. Discriminant analysis has allowed discrimination between 69.6 and 73.9% of the PDD group and 84.6% of the PDD+MD group, indicating robust discriminatory power of the PAI scales and subscales. It also discriminates between 84.2 and 94.7% of PDD according to the criteria that we established in PAI scales and subscales and 100% in PDD+MD, also showing a high level of precision according to the proposed criteria based on the analyses. We have been able to propose a psychometric profile using a set of PAI scales that enables the diagnosis of PDD with and without MD.

It is well established that FM patients present heterogeneous symptomatology not only in moods but in physical symptoms ([36]; [57]), as we can see in our sample, where fatigue, headache, morning tiredness, and stiffness are significantly higher in groups with affective disorders than in NAD; this is unlike pain that remains equally severe across the three FM groups. These findings suggest that affective disorders might not directly cause pain but could potentially regulate it, as there may be a type of fibromyalgia where pain and fatigue are significant entities, with the latter potentially independent of the pain. Notably, the somatization scale has shown high scores in PAI across all three groups, reinforcing the hypothesis of somatic involvement in FM, as FM patients exhibit a higher prevalence of somatic complaints compared to other chronic pain patients. In other words, the pain and somatization problems are the disease themselves, and only somatosensory amplification can predict FM, suggesting that somatization disorder and FM are distinct entities ([16]; [12]).

Dysthymia, characterized by persistent depressive symptoms lasting for at least two years, is often overlooked in medical settings and involves 53–54.5% of our sample, like 50% found in ([17]). Accurately distinguishing between depressed mood and clinical depression is crucial, especially in chronic pain treatment, where PDD is often dismissed. In this study, PAI has demonstrated strong consistency in detecting PDD with high scores in cognitive and emotional depression. We also found low scores in identity alteration, dominance, and grandeur associated with low self-esteem and the belief that one is incapable, dependent on others, and has a profile of low dominance and self-control. Lower self-esteem reduces neurocognitive performance in FM, particularly affecting attention, memory, and planning, which are symptoms of FM. Proper diagnostic methods and criteria help differentiate PDD subtypes, such as ’anxious dysthymia’, characterized by low self-efficacy and anxiety, and ’anergic dysthymia’, characterized by low energy and anhedonia, adding variability to FM groups ([56]; [25]; [43]).

Affective disorders are prevalent in FM and significantly impact quality of life and disability ([14]), often co-occurring with PDD. FM patients usually feel isolated, misunderstood, or rejected by relatives, friends, health workers, and society in general. This may contribute to the high prevalence of depression that is associated with increased pain intensity, irritability, physical and mental strain, functional limitations, the number of tender points, non-restorative sleep, neurocognitive deficits, and fatigue ([16]). In this context, our analysis using PAI scores indicated that detecting major depression requires high scores in cognitive, emotional, and physiological depression; identity alteration; cognitive anxiety; and suicidal ideation; as well as low scores in dominance and grandeur. Additionally, chronic stressors, particularly somatic conditions, can worsen the condition and increase suicide rates. Furthermore, studies indicate a higher prevalence of suicidality in FM patients than in the general population, where their risk of suicide is similar to that observed in other chronic diseases and correlates with depression, anxiety, sleep quality, and overall mental health ([51]).

The findings underscore the relevance of psychological models (e.g., Biopsychosocial, Cognitive Behavioral, and Social Cognitive) in understanding the interaction between chronic pain, mental health, and personality traits in fibromyalgia. These frameworks support the development of tailored interventions, such as cognitive–behavioral strategies, self-efficacy enhancement, acceptance and commitment therapy, and somatic symptom management. Incorporating a biopsychosocial approach in primary care can promote multidisciplinary collaboration, improve patient outcomes, and inform health policy.

Furthermore, the study highlights FM’s complexity, demanding multidimensional assessment and treatment. While both the MCMI-III and the PAI are crucial in diagnosing psychopathology, including chronic pain patients, they differ in their approach. The MCMI-III aids in clinical and personality disorder assessment, providing a categorical dysthymia diagnosis cut-off, while the PAI offers more comprehensive evaluation using dimensional functions to approach MCMI-III diagnoses, potentially enhancing symptom understanding. Among these analyses with PAI scores, results for PDD diagnosis and co-occurrence with major depression episodes entails a cut-off score of 60 in the mentioned scales and subscales. Following group comparison, it was found that 72.2% of patients with affective disorder by PAI were correctly classified in the MCMI-III group with affective disorder, along with 70% of NAD patients, aiding in reducing false positives and improving classification accuracy.

The study has limitations to consider for future research improvements. In this study, a cut-off score of 60 has been used to determine a high score, but using a cut-off of 70 for cognitive and emotional depression will improve the detection of dysthymia with major depression disorder, where we have observed potential accuracy enhancement. A general study limitation may be that despite strict patient recruitment criteria, our cohort might exclude very vulnerable individuals unwilling or unable to participate in tests on a specific day, limiting our sample despite thorough selection. Additionally, we do not have a control group, and we have not conducted a diagnostic interview for dysthymia; instead, we form the groups based on the MCMI-III scores. In future research we want to replicate the study with a PDD group with and without FM. Another limitation is the lack of exploration of normal personality traits, such as those assessed by the Big Five, which could offer further insights into fibromyalgia and related disorders ([27]). Lastly, medication may interfere with present symptomatology, as significant group differences existed and remained unchanged during evaluation, thus not altering the established pattern in patients.

In conclusion, this study provides a theoretical contribution by offering a psychometric profile by PAI that allows for the detection of persistent depressive disorder (PDD), with or without major depression. While the PAI can already assess 25 DSM-based mental disorders, it did not previously include a specific profile for PDD. Our findings help define when a patient may present with dysthymia, contributing to its clinical utility. It is interesting to note that there are few questionnaires that work properly with FM pathology, and it is of clinical interest to be able to conduct a correct assessment that facilitates the detection of affective disorder types and allows for improving prognostic and therapeutic guidance.

## Figures and Tables

**Table 1 ejihpe-15-00002-t001:** Clinical and sociodemographic characteristics of the female sample [N = 66).

CHARACTERISTICS	STATISTICS DESCRIPTIVES
Age (years) (mean [SD])	49.18 [8.09]
Tender points [0–18] (mean [SD])	17.27 [1.35]
Years from FM diagnostic (mean [SD])	6.76 [6.86]
Level of studies (n [%])	
Primary studies	7 [10.6]
Secondary studies	8 [12.1]
Bachelor	12 [18.2]
Professional studies	19 [28.8]
University	20 [30.3]
Associated symptoms *: (mean [SD])	
Morning tiredness	76.44 [20.72]
Unrefreshed sleep	74.88 [19.0]
Fragmented sleep	59.85 [33.07]
Fatigue	77.64 [14.91]
Morning stiffness	71.20 [24.71]
Stiffness after resting	60.91 [26.29]
Subjective swelling	50.42 [32.91]
Paraesthesias	58.45 [27.40]
Headache	60.15 [31.4]
Symptoms of irritable bowel	42.95 [37.66]
Depression symptoms	58.67 [31.51]
Anxiety symptoms	59.92 [35.06]
Subjective difficulties of attention and concentration	65.50 [25.86]
Subjective memory complains	65.80 [24.69]
FIQ **: global score (mean [SD])	67.01 [13.09]
Physical dysfunction	5.88 [2.18]
General discomfort	7.86 [2.59]
Sick leave caused by FM	4.45 [3.45]
Pain at work	6.98 [1.94]
Pain	7.20 [1.56]
Fatigue	7.85 [1.36]
Morning tiredness	7.48 [2.03]
Stiffness	6.85 [2.33]
Anxiety	6.48 [2.70]
Depression	5.63 [2.85]
Stable medication regime (n [%])	
Analgesic (NSAIDs and/or opioids)	43 [68.3]
Anti-inflammatory	37 [58.7]
Antidepressant	47 [74.6]
Type of antidepressant	
ISRS	20 [31.7]
Dual	11 [17.5]
Tricyclic	17 [27]
Benzodiazepine	23 [36.5]
Type of benzodiazepine	
Short life	10 [15.9]
Medium life	3 [4.8]
Long life	10 [15.9]

***Abbreviations:* FIQ:** Fibromyalgia Impact Questionnaire, **FM:** fibromyalgia, ***Note:***
***** Assessment according to a visual analog scale (VAS), maximum score 100; ****** Fibromyalgia Impact Questionnaire [FIQ], maximum score 100; scale maximum score 10.

**Table 2 ejihpe-15-00002-t002:** Clinical and sociodemographic characteristics of the three groups of fibromyalgia patients based on the presence or absence and type of affective disorder.

CHARACTERISTICS	GROUP NAD (n = 30)	GROUP PDD (n = 23)	GROUP PDD+MD (n = 13)	P (Finner’s Correction)	Pairwise
Age (years) (mean [SD])	48.57 [8.67]	50.26 [7.28]	48.69 [8.48]	0.786	
Tender points (0–18) (mean [SD])	17.17 [1.46]	17.39 [0.94]	17.27 [1.84]	0.819	
Years from diagnostic (mean [SD])	7.97 [7.47]	5.76 [6.18]	5.74 [6.58]	0.461	
Level of studies (n [%])				0.747	
Primary studies	2 [6.7]	4 [17.4]	1 [7.7]		
Secondary studies	4 [13.3]	1 [4.3]	3 [23.1]		
Bachelor	5 [16.7]	5 [21.7]	2 [15.4]		
Professional studies	8 [26.7]	8 [34.8]	3 [23.1]		
University	11 [36.7]	5 [21.7]	4 [30.8]		
Associated symptoms * (mean [SD])					
Morning tiredness	73.20 [22.04]	74.30 [21.22]	87.69 [12.35]	0.148	
Unrefreshed sleep	70.40 [22.32]	77.61 [12.51]	80.38 [19.19]	0.345	
Fragmented sleep	54.67 [35.01]	63.26 [26.65]	65.77 [39.15]	0.432	
Fatigue	75.5 [15.72]	74.09 [13.36]	88.85 [10.43]	**0.034**	c
Morning stiffness	66.8 [27.41]	59.13 [28.55]	79.62 [9.23]	0.053	
Stiffness after resting	54.17 [26.26]	53.17 [32.32]	79.62 [9.23]	**0.017**	c
Subjective swelling	51.5 [30.77]	52.00 [30.64]	43.08 [39.87]	0.870	
Paraesthesias	58.9 [26.96]	64.09 [30.66]	68.85 [19.80]	0.426	
Headache	51.2 [32.29]	64.09 [30.66]	73.85 [25.75]	0.080	
Symptoms of irritable bowel	34 [33.45]	50.65 [38.56]	50.00 [43.39]	0.354	
Depression symptoms	44.17 [34.19]	64.65 [23.09]	81.54 [20.35]	**0.012**	c
Anxiety symptoms	45.17 [38.56]	61.52 [27.23]	91.15 [10.03]	**<0.005**	c
Subjective difficulties of attention and concentration	62.83 [25.28]	66.87 [26.24]	69.23 [27.90]	0.574	
Subjective memory complains	65 [22.28]	72.09 [29.69]	56.54 [34.96]	0.403	
FIQ **: global score (mean [SD])	62.20 [12.27]	67.86 [12.54]	75.23 [11.13]	**0.028**	b
Physical dysfunction	5.56 [1.99]	6.14 [2.26]	6.16 [2.49]	0.462	
General discomfort	7.74 [2.74]	7.88 [2.93]	8.60 [1.38]	0.732	
Sick leave caused by FM	4.49 [3.7]	4.03 [3.00]	5.10 [3.81]	0.062	
Pain at work	6.55 [2.01]	7.26 [2.00]	7.46 [1.56]	0.034	
Pain	7.03 [1.59]	7.13 [1.66]	7.69 [1.31]	0.732	
Fatigue	7.31 [1.44]	8.17 [1.19]	8.46 [1.05]	0.062	
Morning tiredness	6.76 [2.06]	7.65 [1.96]	8.77 [1.36]	**0.034**	b
Stiffness	6.31 [2.48]	6.78 [2.15]	8.15 [1.90]	0.141	
Anxiety	5.86 [2.70]	6.65 [2.49]	7.54 [2.87]	0.183	
Depression	4.59 [3.01]	5.87 [2.13]	7.54 [2.69]	**0.034**	b, c
Stable medication regime (n [%])					
Analgesic (NSAIDs and/or opioids)	18 [62.1]	15 [65.2]	10 [90.9]	0.354	
Anti-inflammatory	17 [58.6]	13 [56.5]	7 [63.6]	0.925	
Antidepressant	17 [58.6]	20 [87.0]	10 [90.9]	**<0.005**	a
Type of antidepressant				0.107	
ISRS	10 [34.5]	9 [39.1]	1 [9.1]		
Dual	4 [13.8]	3 [13.0]	4 [36.4]		
Tricyclic	4 [13.8]	8 [34.8]	5 [45.5]		
Benzodiazepine	12 [41.4]	6 [26.1]	5 [45.5]	0.529	
Type of benzodiazepine				0.529	
Short life	5 [17.2]	2 [8.7]	3 [54.5]		
Medium life	3 [10.3]	0 [0.0]	3 [27.3]		
Long life	4 [13.8]	4 [17.4]	2 [18.2]		

***Abbreviations:* NAD:** patients without affective disorder; **PDD:** patients with persistent depressive disorder; **PDD+DM:** patients with persistent depressive disorder and major depression; **FIQ:** Fibromyalgia Impact Questionnaire. ***** Assessment according to a visual analog scale (VAS), maximum score 100; ****** Fibromyalgia Impact Questionnaire (FIQ), maximum score 100; scale maximum score 10. (a) NAD vs. PDD 2; (b) NAD vs. PDD+MD; (c) PDD vs. PDD+MD.

**Table 3 ejihpe-15-00002-t003:** Pairwise comparison in PAI scales between groups of study by Kruskal–Wallis and Dunn’s non-parametric comparison.

PAI Clinical Scale	P (Finner’s Correction)	NAD vs. PDD P (Cohen’s d)	NAD vs. PDD+MD P (Cohen’s d)	PDD vs. PDD+MD P (Cohen’s d)
Anxiety	0.135	1.00 (0.26)	**0.030 (1.03)**	0.190 (0.74)
Depression	**<0.005**	**0.024 (0.90)**	**<0.005 (1.85)**	**0.027 (1.28)**
Schizophrenia	0.074	0.812 (0.38)	**0.010 (1.25)**	0.161 (0.61)
Suicidal ideation	0.104	0.619 (0.33)	**0.018 (0.94)**	0.312 (0.56)
Dominance	0.104	**0.017 (0.81)**	0.404 (0.28)	1.00 (0.62)
PAI Clinical Subscale				
Illness-health concern	0.113	1.00 (0.19)	**0.021 (0.85)**	0.203 (0.67)
Cognitive anxiety	**0.047**	0.516 (0.44)	**0.004 (1.18)**	**0.136 (0.78)**
Physiological anxiety	0.135	1.00 (0.25)	**0.032 (1.00)**	0.161 (0.61)
Cognitive depression	**<0.005**	**0.015 (0.93)**	**0.000 (1.64)**	**0.084 (0.75)**
Emotional depression	**<0.005**	**0.050 (0.79)**	**0.000 (1.59)**	**0.132 (0.96)**
Physiological depression	0.074	0.653 (0.42)	**0.009 (1.09)**	**0.190 (0.71)**
Grandeur	**0.047**	**0.052 (0.79)**	**0.011 (0.90)**	1.00 (0.26)
Identity alteration	**0.014**	**0.091 (0.69)**	**0.000 (1.51)**	**0.168 (0.74)**
PAI Complementary items				
Potential suicide index	**0.014**	0.489 (0.46)	**0.000 (1.59)**	**0.030 (0.84)**
Potential violence index	0.158	1.00 (0.14)	0.088 (0.65)	**0.060 (0.84)**
Treatment difficulty index	**0.047**	0.546 (0.17)	0.069 (0.68)	**0.004 (0.78)**

***Abbreviations:* PAI:** Personality Assessment Inventory; **NAD:** patients without affective disorder; **PDD:** patients with persistent depressive disorder; **PDD+DM:** patients with persistent depressive disorder and major depression. Measures that are significant in the discriminant analysis are shown in bold.

**Table 4 ejihpe-15-00002-t004:** Standardized coefficients and canonical correlation values of discriminant analysis.

PAI Clinical Scales	PDD vs. NAD	PDD vs. PDD+MD
Dominance	**−0.266**	-
PAI Clinical Subscales		
Cognitive depression	**0.303**	0.107
Emotional depression	**0.176**	**0.682**
Physiological depression	-	**0.545**
Cognitive anxiety	-	0.414
Grandeur	−0.415	-
Identity alteration	0.370	0.281
PAI Complementary items		
Potential suicide index	-	−0.204
Discriminant analysis indices		
Eigenvalue	0.379	0.536
Significance	0.008	0.038
Canonical correlation	0.424	0.591

***Abbreviations:* NAD:** patients without affective disorder; **PDD:** patients with persistent depressive disorder; **PDD+DM:** patients with persistent depressive disorder and major depression. The values of the measures with the highest coefficients are shown in bold.

## Data Availability

The data presented in this study are available on request from the corresponding author due to privacy restrictions.

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
