# Peer review of "Outlining the Psychological Profile of Persistent Depression in Fibromyalgia Patients Through Personality Assessment Inventory (PAI)"

_ejihpe, 2025, doi:10.3390/ejihpe15010002_

Round 1
Reviewer 1 Report
Comments and Suggestions for Authors
25 Nov 2024
ejihpe-3274633 - Outlining the Psychological Profile of Persistent Depression in Fibromyalgia Patients through Personality Assessment Inventory (PAI)
Dear authors,
I would like to thank you for the opportunity to read your manuscript. The paper is interesting, well-written, and the statistical analyses are reliable. I do find, however, that the manuscript, in its current version, must be somewhat improved, before could be considered for publication in the European Journal of Investigation in Health, Psychology and Education.
My primary worries stem from the lack of a theoretical framework that may contextualise the study's findings and support its goals.
Introduction: Only the description of clinical problems and a few studies on the relationship between fibromyalgia and anxiety and depressive disorders are covered in the Introduction. A number of well-known models in clinical and health psychology, including the Biopsychosocial Model, Cognitive Behavioural Theory, the Social-Cognitive Theory, and the Somatic Symptom Model, can help explain the intricate relationship between chronic pain, mental health, and personality traits. These models look at the chronic pain that characterizes fibromyalgia as being determined by conditions of a psychological nature, also having consequences at this level in the lives of patients, as the study authors emphasize.
Discussion: Furthermore, these models will enable the development of the study's practical consequences within the framework of psychological assessment and intervention. Given the soundness of the findings, recommendations for psychological interventions for fibromyalgia patients, health policy, and primary healthcare may be discussed.
Author Response
Introduction: Thank you for this insightful suggestion. We agree that including theoretical models from clinical and health psychology enriches the understanding of the relationship between chronic pain, mental health, and personality traits in fibromyalgia. In response, we have added a paragraph in the Introduction (line 50-60) that briefly outlines the relevance of models such as the Biopsychosocial Model, Cognitive Behavioural Theory, Social-Cognitive Theory, and the Somatic Symptom Model.
Discussion: Thank you for your valuable suggestion. We agree that linking the findings to practical applications within psychological assessment and intervention is important. To address this, we have included a paragraph in the Discussion (384-390) highlighting how these theoretical models can inform tailored interventions, such as cognitive-behavioural strategies, self-efficacy enhancement, and somatic symptom management. We also discuss the potential for a biopsychosocial approach to foster multidisciplinary collaboration, improve patient outcomes, and guide health policy development.

Reviewer 2 Report
Comments and Suggestions for Authors
Thank you for submitting your paper. I found it to be a highly valuable and important contribution to the field. I also felt that I was able to fully understand the areas of focus. With that in mind, I would like to offer the following comments, as I believe the points below may pose significant issues for this study as a research paper.
Given the statistical analyses planned, do you have any concerns about the sample size? Is there a risk of insufficient statistical power?
In this study, I believe the issue of multiple testing may be significant. What are your thoughts on this?
Minor Comment
In the fifth paragraph of the introduction, the limitations and challenges of PAI are discussed, but please provide a more detailed explanation in this section.
Author Response
Given the statistical analyses planned, do you have any concerns about the sample size? Is there a risk of insufficient statistical power?Thank you for raising this important point. We agree that having a larger sample size would have been desirable to increase statistical power. This limitation is acknowledged and discussed in the manuscript as a potential constraint of the study. To address the limitations posed by the sample size, we placed particular emphasis on determining and reporting effect sizes, which mitigate the impact of sample size on the interpretation of results. Additionally, the sample size was taken into consideration when conducting the analyses, as shown in Table 3.
In this study, I believe the issue of multiple testing may be significant. What are your thoughts on this? Thank you for pointing out the potential issue of multiple testing. In our analyses, we used the non-parametric Kruskal-Wallis test to compare the three groups (NAD, PDD, and PDD+MD) for the PAI clinical scales and subscales, with all reported p-values being two-tailed. To address the risk of inflated Type I error due to multiple pairwise comparisons, we applied Dunn's non-parametric pairwise test, which includes adjustments for multiple comparisons. Additionally, to improve our analysis based on your comment, we applied the Finner correction, which is reflected in Tables 2 and 3 as well as in the results. This approach ensures that the results are robust while minimizing the likelihood of spurious findings. Minor Comment In the fifth paragraph of the introduction, the limitations and challenges of PAI are discussed, but please provide a more detailed explanation in this section. Thank you for your observation. In the Introduction, we have clarified how several PAI diagnoses are related to affective disorders but noted that none of them allow for a diagnosis compatible with persistent depressive disorder (PDD). In the Discussion, we highlight that this study provides a proposal for which PAI scales and subscales define the psychometric profile of PDD, with or without major depression. These additions aim to better explain the link between the PAI and PDD. You can find the revisions in the Introduction (line 107-110) and Discussion (line 342-343 and 418-422).
Reviewer 3 Report
Comments and Suggestions for Authors
I have read this paper with interests. This paper is generally well-written. However, it remains unclear to me why the authors wish to use PAI to detect PDD, given there are other specialized tools for detecting PDD.
1. One limitation of this research is that it fails to use other personality inventories such as the Big Five, which should be discussed in the limitation section (https://doi.org/10.1016/j.comppsych.2024.152514).
2. What are the theoretical contribution of this study?
3. It remains unclear to me why you wish to use PAI to detect PDD, because there are a lot of already developed and validated inventories specialized in detecting PDD. Do the authors wish to look at the associations between PAI and PDD?
4. More explanations are needed regarding why PAI is linked to PDD.
Author Response
1. One limitation of this research is that it fails to use other personality inventories such as the Big Five, which should be discussed in the limitation section (https://doi.org/10.1016/j.comppsych.2024.152514).Thank you for this insightful observation. We acknowledge that the inclusion of other personality inventories, such as the Big Five, could have provided additional perspectives on normal personality traits in this study. The main goal of our study is to further our knowledge in the psychopathological diagnosis through PAI. We have added this point as a limitation in the manuscript, highlighting the potential benefits of integrating alternative inventories in future research. You will find this addition in the Limitations section (412-415).
2. What are the theoretical contribution of this study?Thank you for these questions. The theoretical contribution of this study lies in providing a psychometric profile for the PAI that allows for the detection of persistent depressive disorder (PDD), with or without major depression, a condition not previously assessed with this instrument. While there are existing inventories for detecting PDD, our study aims to extend the clinical utility of the PAI, which already evaluates 25 DSM-based disorders, by offering a profile to identify dysthymia. This addresses a gap in the instrument and enhances its application in clinical contexts. We have added this explanation at the end of the first paragraph of the Discussion (line 342-343).
3. It remains unclear to me why you wish to use PAI to detect PDD, because there are a lot of already developed and validated inventories specialized in detecting PDD. Do the authors wish to look at the associations between PAI and PDD? 4. More explanations are needed regarding why PAI is linked to PDD. Thank you for your observation. In the Introduction, we have clarified how several PAI diagnoses are related to affective disorders but noted that none of them allow for a diagnosis compatible with persistent depressive disorder (PDD). In the Discussion, we highlight that this study provides a proposal for which PAI scales and subscales define the psychometric profile of PDD, with or without major depression. These additions aim to better explain the link between the PAI and PDD. You can find the revisions in the Introduction (line 107-110) and Discussion (line 342-343 and 418-422).
Reviewer 4 Report
Comments and Suggestions for Authors Thanks for the opportunity to review this paper. It is very well written, and quite interesting for advancing in the field. So, I only have minor comments: - Was the project submitted to any ethical committee? There is no information about ethics procedures. - Results and statistical procedures are quite fascinating and sophisticated. However, do the authors estimated a minimum number of participants needed for doing such analyses comparing different subgroups? In any case, probably the small sample size should be discussed in detail as a limitation in the discussion section. - This assumption in the discussion "These findings suggest that affective disorders may not directly cause pain but can modulate it as we can find a type of fibromyalgia where pain and fatigue are significant entities, with the latter being independent of the pain" is too strong. Authors can not be sure about that because they do not have a causal design for their study. - I would discuss I little more about clinical implications and future research.
Author Response
Was the project submitted to any ethical committee? There is no information about ethics procedures.
Thank you for your observation. The ethical approval information is included at the end of the manuscript under the section Institutional Review Board Statement. Specifically, this research followed the guidelines set forth in the Declaration of Helsinki and obtained approval from the Ethical Committees of Parc de Salut Mar of Barcelona (reference 6932/I) and the Commission on Ethics in Animal and Human Experimentation (CEEAH) at the Autonomous University of Barcelona (UAB) (reference 6496). These approvals ensure that the study complies with the ethical standards of both hospital and university settings.
- Results and statistical procedures are quite fascinating and sophisticated. However, do the authors estimated a minimum number of participants needed for doing such analyses comparing different subgroups? In any case, probably the small sample size should be discussed in detail as a limitation in the discussion section.
Thank you for your comment and for highlighting this important aspect. This study used a consecutive non-probabilistic sampling approach based on strict inclusion and exclusion criteria. Due to the nature of this sampling design, we did not estimate a minimum sample size prior to conducting the analyses. However, we acknowledge the limitations of the sample size and have addressed this as a study limitation in the Discussion section (line 407-410)
This assumption in the discussion "These findings suggest that affective disorders may not directly cause pain but can modulate it as we can find a type of fibromyalgia where pain and fatigue are significant entities, with the latter being independent of the pain" is too strong. Authors can not be sure about that because they do not have a causal design for their study.
Thank you for highlighting this concern. We have revised the phrasing in the Discussion section to make it more conditional and reflective of the study's observational nature. The updated sentence will appear in line 348-350.
I would discuss I little more about clinical implications and future research.
Thank you for your suggestion. We have expanded the Discussion section to address clinical implications and future research directions. Specifically, we discuss how different types of interventions could benefit from the integration of psychological models (e.g., Biopsychosocial and Cognitive Behavioural) to better address both physical and psychological symptoms in fibromyalgia. (line to be confirmed). Regarding future research, we propose expanding the sample size and exploring comparisons between patients with fibromyalgia and dysthymia versus dysthymic patients without fibromyalgia. This could provide further insight into the interaction between chronic pain and affective disorders. You will find these additions in line 418-422 and 412-415.

Round 2
Reviewer 2 Report
Comments and Suggestions for Authors
Thank you for the revisions to your manuscript. I would also like to express my gratitude to the authors for their efforts in thoroughly addressing the reviewers' comments.
With the current revisions, the authors have succeeded in clarifying the study's background and strengthening the robustness of the results. I look forward to seeing further advancements in this line of research, including studies with larger sample sizes and insights derived from multivariate analyses in the future.
Reviewer 3 Report
Comments and Suggestions for Authors
Thank you for addressing my comments.